# TIDALDECODE: FAST AND ACCURATE LLM DECODING WITH POSITION PERSISTENT SPARSE ATTENTION

**Lijie Yang**[*]
Carnegie Mellon University
lijiey@andrew.cmu.edu

**Zhihao Zhang**[*]
Carnegie Mellon University
zhihaoz3@cs.cmu.edu

**Zhuofu Chen**
Carnegie Mellon University
zhuofuc@cs.cmu.edu

**Zikun Li**
Carnegie Mellon University
zikunl@cs.cmu.edu

**Zhihao Jia**
Carnegie Mellon University
zhihao@cmu.edu

## ABSTRACT

Large language models (LLMs) have driven significant advancements across diverse NLP tasks, with long-context models gaining prominence for handling extended inputs. However, the expanding key-value (KV) cache size required by Transformer architectures intensifies the memory constraints, particularly during the decoding phase, creating a significant bottleneck. Existing sparse attention mechanisms designed to address this bottleneck have two limitations: (1) they often fail to reliably identify the most relevant tokens for attention, and (2) they overlook the spatial coherence of token selection across consecutive Transformer layers, which can lead to performance degradation and substantial overhead in token selection. This paper introduces TidalDecode, a simple yet effective algorithm and system for fast and accurate LLM decoding through position persistent sparse attention. TidalDecode leverages the spatial coherence of tokens selected by existing sparse attention methods and introduces a few token selection layers that perform full attention to identify the tokens with the highest attention scores, while all other layers perform sparse attention with the pre-selected tokens. This design enables TidalDecode to substantially reduce the overhead of token selection for sparse attention without sacrificing the quality of the generated results. Evaluation on a diverse set of LLMs and tasks shows that TidalDecode closely matches the generative performance of full attention methods while reducing the LLM decoding latency by up to $2.1\times$. Code is available at https://github.com/DerrickYLJ/TidalDecode.

## 1 INTRODUCTION

Large language models (LLMs) have revolutionized natural language processing (NLP) by achieving state-of-the-art performance on various tasks, including machine translation, summarization, and question-answering. Their ability to generate human-like text has made them indispensable across various domains, from academic research to industry applications (Touvron et al., 2023). As LLMs expand to handle more extended contexts, their potential applications increase dramatically. Long-context processing is vital for tasks such as chain-of-thought reasoning (Wei et al., 2023), document summarization (Huang et al., 2021), and retrieval-augmented generation (Ram et al., 2023; Zhang et al., 2024b). However, extending context length further exacerbates the memory and compute bottlenecks inherent in the Transformer architectures.

LLM inference includes two stages: prefilling and decoding. The prefilling stage computes the activations for all input tokens and stores the keys and values for all tokens in the KV cache, allowing the LLM to reuse these keys and values to compute attention for future tokens. In each decoding stage, the LLM decodes one new token using all input tokens and previously generated tokens. The KV cache size grows linearly in the sequence length (Kwon et al., 2023). For instance, with a context length of 128K tokens, the KV cache of LLama2-7B with half-precision can easily reach 64 GB[1],

---

[*]Equal contribution

[1]The KV cache size is computed as: Layers$\times$KV Heads$\times$Head Dim$\times$Seq Len$\times$FP16 Size$\times$2 (for K+V) $= 32 \times 32 \times 128 \times 128K \times 2$ bytes $\times 2 = 64$ GB.

creating substantial memory pressure for LLM serving. In addition, the LLM decoding stage is memory-bounded since decoding one new token requires accessing all previous tokens in the KV cache, making KV cache access the primary bottleneck for LLM decoding. This memory-bound nature severely limits the scalability and efficiency of LLM serving.

To address the memory bottleneck for KV cache access, recent work has introduced sparse attention, which approximates full attention using a small portion of tokens with high attention scores. Compared to full attention, sparse attention reduces computation cost and memory access while preserving the LLM's generative performance (Ge et al., 2024; Zhang et al., 2023). Existing sparse attention techniques can be classified into two categories: eviction- and selection-based methods. First, eviction-based sparse attention methods reduce memory usage for the KV cache by selectively discarding less relevant tokens from the KV cache, therefore reducing the number of tokens computed in attention mechanisms (Xiao et al., 2023; Zhang et al., 2023). While these methods decrease the size of the KV cache, they can be inadequate for tasks where critical information is carried by tokens that are prematurely evicted, such as the needle-in-the-haystack tasks. On the other hand, selection-based sparse attention methods maintain all tokens in the KV cache, estimate their attention scores, and select a small subset of tokens to participate in each LLM decoding step. While these selection-based methods introduce various techniques to pinpoint relevant tokens, they often ignore the correlation of token selection across consecutive Transformer layers and introduce additional overhead for attention score estimation. Moreover, these approaches are prone to distribution shifts caused by appending sparsely attended, biased KV representations back into the cache.

This paper presents TidalDecode, a streamlined and efficient framework designed for fast and accurate LLM decoding, utilizing position persistent sparse attention (PPSA). A key insight behind TidalDecode is the observation that tokens selected for sparse attention — based on their highest attention scores — exhibit significant overlap across consecutive Transformer layers within each decoding phase, which we refer to as spatial coherence. Instead of selecting tokens for sparse attention independently in each layer, TidalDecode introduces a few token selection layers performing full attention to identify the tokens with the highest attention scores, which are shared across the following sparse attention layers, reducing the overhead of token selection. Empirical evidence shows that using just two token selection layers — one at the beginning and one in the middle — is sufficient to achieve high generative performance while minimizing computation and memory overheads. Additionally, to address the KV cache distribution shift, TidalDecode introduces a cache-correction mechanism that periodically refills the KV cache using full attention for all sparsely decoded tokens to mitigate bias in the KV representations.

Comprehensive evaluation with the LongChat-7b-v1.5-32k, Llama-3-8B, Llama-3-70B, and Llama-3.1-8B models on the Needle-in-the-Haystack, PG-19, and LongBench tasks demonstrates that TidalDecode can consistently achieve the best performance efficiency trade-off compared with the best existing sparse attention methods. We have implemented custom GPU kernels for PPSA and an end-to-end system for TidalDecode. Compared with existing full and sparse attention implementations, our system reduced the end-to-end inference latency by up to $2.1\times$ and $1.2\times$, respectively. In conclusion, our contributions are:

- We propose TidalDecode, a streamlined and efficient algorithm and system for fast and high-quality LLM decoding, utilizing position persistent sparse attention.

- To address KV cache distribution shifts, we introduce a cache-correction mechanism that periodically refills the KV cache with using full attention for sparsely decoded tokens.

- Empirically, we demonstrate the effectiveness and efficiency of TidalDecode through comprehensive evaluation, showing that TidalDecode significantly outperforms existing sparse attention methods.

## 2 RELATED WORK

**Long-context model.** Efficiently handling long-context inputs is essential for various LLM tasks in real-world applications such as document summarization, question answering, and dialogue systems (Wang et al., 2024). Recent advancements, including rotary positional encoding (RoPE) (Su et al., 2023), have enabled models to manage extended context lengths effectively. The Llama-3 model series supports up to 8K tokens, with enhanced versions such as Gradient-AI-Llama3 (AI, 2024a) and Llama 3.1 (AI, 2024b) extending this limit to 128K tokens. Additionally, proprietary LLMs such

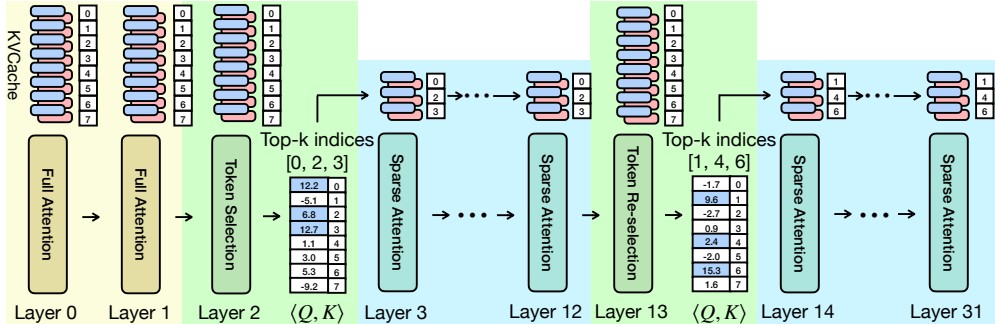

Figure 1: An overview of the decoding step in TidalDecode, which performs full attention for the first two layers, full attention with token selection for the third layer and a middle layer, and position persistent sparse attention for all other layers.

as GPT-4 Turbo and GPT-4o (OpenAI, 2024) support up to 128K tokens, and Claude 3.5 Sonnet allows up to 200K tokens (Anthropic, 2024). While recent work has introduced efficient attention kernel implementation (Dao et al., 2022; Dao, 2023), processing long-context inputs continues to be constrained by significant memory usage and computational costs from the extended KV cache. TidalDecode is designed to mitigate these challenges by reducing latency and memory overhead through an efficient strategy for selecting tokens with the highest attention scores and one-time intermediate re-calibration, ensuring both efficiency and high-quality output.

To alleviate the intrinsic computational and memory bottleneck in long-context LLM inference, recent works on sparse attention have approached this problem from two main perspectives: eviction- and selection-based methods.

**Eviction-based sparse attention.** Xiao et al. (2023); Zhang et al. (2024a) propose to reduce KV cache memory usage by evicting tokens that are considered less relevant during inference. These suffer from potential performance degradation, especially in tasks where every token may carry crucial information (e.g., needle-in-the-haystack tasks), since tokens with high importance for a future decoding step can be mistakenly evicted as the generation proceeds, which makes selection-based methods more popular choices in latest sparse attention works.

**Selection-based sparse attention.** Instead of evicting past tokens in the KV cache, Child et al. (2019); Kitaev et al. (2020); Choromanski et al. (2020); Tang et al. (2024); Ribar et al. (2023) preserve the full KV cache and only select important tokens to attend with the attention module on the fly. More specifically, Child et al. (2019) leverages a fixed attention mask to select tokens while Tang et al. (2024); Ribar et al. (2023); Choromanski et al. (2020); Kitaev et al. (2020) aim to identify and retain the most relevant tokens at each layer by approximating attention scores. Although these methods are more selective, they operate independently at each layer and are not guaranteed to obtain the ground-truth tokens with the highest attention scores, failing to capture token relevance patterns that persist across layers. Moreover, attention score estimation algorithms sometimes introduce unnecessary complexity, diminishing the practical efficiency gains they are designed to achieve. Improving upon prior works, TidalDecode leverages a shared pattern of most important tokens across consecutive layers to further reduce the computational overhead and memory access required for token selection.

## 3 METHODOLOGY

This section introduces TidalDecode, an efficient algorithm and system for fast LLM decoding using position persistent sparse attention and KV cache correction. Figure 1 shows an overview of TidalDecode. TidalDecode uses the same prefilling mechanism as existing systems and performs full attention to compute the key-value (KV) cache for all prompt tokens. In each decoding step, TidalDecode uses three types of attention layers: full attention, full attention with token selection, and position persistent sparse attention. First, TidalDecode performs full attention for the initial Transformer layers to avoid early performance degradation as identified by prior work (Tang et al., 2024). Second, the layer immediately after full attention and a single middle layer (e.g., layer 2 and 13 in Figure 1) perform full attention with token selection, where TidalDecode stores the inner

product[2] between the current query and key vectors of all tokens in KV cache during full attention and then selects $k$ tokens contributing to the highest attention scores. Third, all other layers perform position persistent sparse attention, where only tokens selected from the previous token selection layer are loaded from the KV cache to perform attention computation.

## 3.1 POSITION PERSISTENT SPARSE ATTENTION (PPSA)

Attention mechanisms have been widely used in today's LLMs. For each attention head, the output is computed via scaled multiplicative formulation as follows.

$$A_i = Q_i K_i / \sqrt{d}, \quad H_i = \text{softmax}(A_i) V_i \tag{1}$$

where $Q_i$, $K_i$, and $V_i$ are the query, key, and value tensors for the $i$-th attention head. $A_i$ is a matrix representing the attention scores between tokens, and $H_i$ is the output of the $i$-th attention head. Instead of attending to all input tokens, existing sparse attention methods approximate attention computation by attending the query $Q_i$ to a subset of previous tokens with the highest attention scores. Prior work generally performs token selection for individual attention heads and Transformer layers, introducing high runtime overhead. For example, identifying the top-k tokens with the highest attention scores can introduce additional overhead and even take longer than computing full attention under certain circumstances (see Figure 6), thus diminishing the benefits of performing sparse attention.

The key insight behind TidalDecode's position persistent sparse attention is an observation that tokens with the highest attention scores for consecutive Transformer layers highly overlap. We use the Llama-3-8B model and the needle-in-the-haystack test on the PG-19-mini dataset with a context length of 100K tokens to quantify this observation. We arbitrarily select 100 requests from the dataset, insert needles to random depth, compute full attention, and analyze the correlation of attention scores patterns between different Transformer layers. We then report the head-wise cosine similarity for attention scores between adjacent layers in Figure 2a and the top-256 recall rate for different choices of token re-selection layer in Figure 2b. We observe that with only one token selection layer at the third layer, the average recall rates are less than 20% as shown in Figure 2b. However, as we choose an additional layer 13 to perform re-selection, the average recall rates boost to almost 40% due to the improvement on the token re-selection as shown in Figure 2a, where the token re-selection layer lies in the least correlated regions, identified by the red box.

---

**Algorithm 1** TidalDecode

1: **Input:** Current embedding $h$, KV cache $\mathcal{C}$, token budget $m$
2: **Output:** Logits
3: **Initialize:** $\rho = []$       ▷ Initialize the token buffer to store selected tokens
4: **for** each decoder layer $i$ **do**
5:     $q, k, v = f(W_{qkv}, h)$
6:     $\mathcal{C}.\text{append}(k, v)$
7:     **if** $i$ is Full Attention Layer **then**
8:         $o = \text{FullAttention}(q, \mathcal{C}[:])$     ▷ Dense attention with the full KVCache
9:     **else if** $i$ is Token Selection Layer **then**
10:        $o = \text{FullAttention}(q, \mathcal{C}[:])$     ▷ Dense attention with the full KVCache
11:        $K \leftarrow \mathcal{C}.\text{getKey}, \rho := \text{argTopK}(\langle q, K \rangle, m)$     ▷ Update token buffer
12:     **else**
13:        $o = \text{SparseAttention}(q, \mathcal{C}[\rho])$     ▷ Sparse attention with the tokens in the token buffer
14:     **end if**
15:     $h = \text{FFN}(o)$
16: **end for**
17: $\text{logits} = \text{lm\_head}(h)$
18: **return** logits

---

Based on this observation, we design position persistent sparse attention to maximally leverage the token overlaps between consecutive Transformer layers to reduce the computation cost for token

---

[2]We don't store attention score as state-of-the-art attention kernels don't materialize the attention score. Since the softmax operation is ordering invariant, we store the inner product value instead.

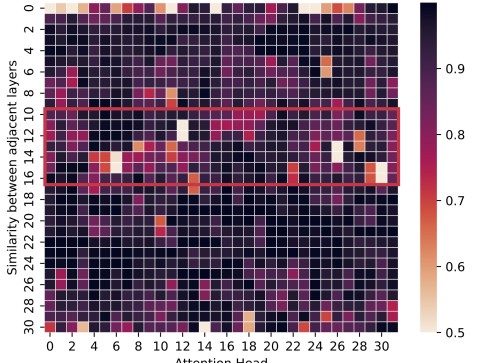
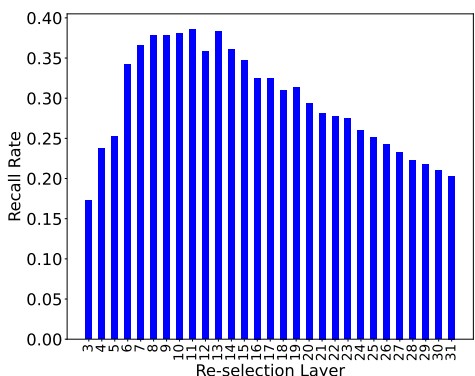

(a) Cosine similarity of the attention scores for adjacent layers. The x-axis is the attention head index, and the y-axis corresponds to the correlation between layer $i$ and layer $i + 1$ for the $i$-th entry.

(b) Recall rate by re-selection Layer, defined as the averaged overlap ratio between the tokens selected by TidalDecode and the ground truth ones across all sparse attention layers.

Figure 2: By analyzing the attention score patterns from a 100K-context-length Needle-in-the-Haystack test conducted on PG-19-mini, 2a shows the attention similarity between adjacent layers, where the least correlated region is identified by the red bounding box. 2b depicts the top-256 recall rates, indicating that different choices of re-selection layers have a high impact on the recall rates — there is a clear peak, delineating the optimal layers for token re-selection.

selection while achieving high predictive performance. Algorithm 1 shows the TidalDecode algorithm for interleaving full attention and PPSA layers. After the initial full attention layers, TidalDecode uses a token selection layer that computes full attention and selects tokens with the highest attention scores. To select tokens, TidalDecode stores the inner product $\langle Q, K \rangle$ on the fly together with full attention calculation. TidalDecode then selects the top $k$ tokens with the highest inner product values to form a token set $\mathcal{T}$. Note that using the inner product to select top-$k$ is equivalent to the post-softmax attention score as the softmax operator is ordering invariant. All PPSA layers after a token selection layer computes sparse attention by only loading the keys and values for tokens in $\mathcal{T}$, thus limiting the number of tokens participating in attention computations and reducing memory access.

A straightforward approach to designing TidalDecode is to select the tokens $\mathcal{T}$ once after full attention and perform PPSA using the same set of tokens for all subsequent layers. However, our preliminary experimentation shows that using a single token set for all Transformer layers reduces the LLM's predictive performance by a large margin since distant Transformer layers are less correlated compared to consecutive layers, as shown in Figure 2. To address this issue, TidalDecode performs token re-selection in a middle layer, where TidalDecode recalibrates the selected tokens with the highest attention scores by applying full attention and re-selecting top-$k$ token to update $\mathcal{T}$, ensuring that token selection remains optimal for the remaining layers. This re-selection mechanism significantly boosts the model performance and promotes accurate and efficient PPSA throughout the model.

Extensive evaluation on both small and large models on a wide range of datasets shows that using a single middle layer for token re-selection is sufficient to preserve the LLM's generative performance, while introducing small runtime overhead. However, deciding which layer to perform token re-selection is critical to model performance. As shown in Figure 2, choosing different layers for token re-selection results in different recall rates, where layer 11 and 13 achieve optimal performance. Introducing a one-time token re-selection at an optimal layer ensures the selected tokens are re-calibrated, effectively mitigating the drift in token importance and elevates accuracy from 15% (without re-selection) to almost 40%.

## 3.2 KV CACHE CORRECTION

For tokens decoded by sparse attention methods, their key/value representations can deviate from the original representation of full attention decoded ones, which we refer to as polluted tokens. The problem can be further exacerbated as their KV pairs are added to the KV cache, resulting in the error accumulation or distribution shift of the KV cache. This can lead to model performance drop in scenarios where the generation length is fairly long. To this end, TidalDecode uses a cache-correction mechanism as shown in Figure 3 to periodically correct the polluted tokens in the KV cache. For

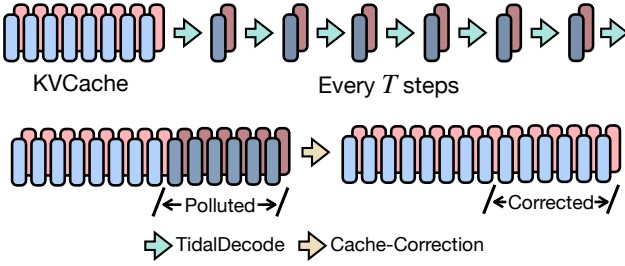

Figure 3: Cache Correction

every $T$ decoding step performed by TidalDecode, there will be a cache correction step through a prefill over all polluted tokens to update their KV representations in the cache. The choice of $T$ can be at the level of thousands of decoding steps but also depend on different models and tasks. Notice that the cache correction step can be performed concurrently with the sparse decoding step. Nevertheless, we haven't used cache correction in our evaluations to make it a fair comparison against existing methods.

## 4 EXPERIMENTS

### 4.1 EXPERIMENT SETTING

In this section, we conduct extensive experiments to assess both the performance and efficiency of TidalDecode. Our evaluations are performed on widely used open-source models, including Llama-2-7B Touvron et al. (2023) and Llama-3-8/70B. Both models are pretrained decoder-only transformers, exhibiting similar yet distinct architectural features. For instance, Llama 3-8B incorporates group query attention (GQA), a feature not present in Llama 2-7B. In Section 4.2, we evaluate TidalDecode's performance on various tasks, including needle-in-the-haystack, language modeling on PG-19, and LongBench. In Section 4.3, we write customized attention kernels and compare TidalDecode's kernel efficiency against existing state-of-the-art sparse attention methods. Finally, in Section 4.4, we conclude our evaluations with a detailed sensitivity analysis on the choice of different token selection layers. We use TD+LX to denote TidalDecode with layer X selected as the token re-selection layer throughout this section.

### 4.2 PERFORMANCE EVALUATION

To evaluate the effectiveness of TidalDecode, we conduct thre key downstream NLP experiments: the needle-in-the-haystack (Peng et al., 2023) test, perplexity evaluation on the PG-19 dataset (Rae et al., 2019), and eight datasets in LongBench (Bai et al., 2023). These tasks provide robust benchmarks for measuring both sparse attention models' ability to retrieve critical information in challenging scenarios and their performance on long-context language modeling tasks.

#### 4.2.1 NEEDLE-IN-THE-HAYSTACK

Table 1: Results of 10k-context-length Needle-in-the-Haystack test on LongChat-7b-v1.5-32k. TidalDecode achieves the same or better results than Quest (Tang et al., 2024) and significantly better results than cache eviction algorithms such as H2O (Zhang et al., 2024a), TOVA (Oren et al., 2024), and StreamingLLM (Xiao et al., 2023). TidalDecode achieves full accuracy with only a 512 token budget.

| Method / Budget | K=32 | K=64 | K=128 | K=256 | K=512 |
|---|---|---|---|---|---|
| H2O | 0% | 1% | 1% | 1% | 3% |
| TOVA | 0% | 1% | 1% | 3% | 8% |
| StreamingLLM | 1% | 1% | 1% | 3% | 5% |
| Quest | 65% | **99%** | **99%** | **99%** | **100%** |
| TD+L7(Ours) | **73%** | 92% | 98% | **99%** | **100%** |

The Needle-in-the-Haystack test assesses LLMs' ability to handle long-dependency tasks, which is particularly critical for sparse attention algorithms. Eviction-based methods (H2O, StreamingLLM)

Table 2: Comprehensive results of 10K-, 32K-, and 100K-context-length Needle-in-the-Haystack test on Llama-3-8B-Instruct-Gradient-1048k, Llama-3.1-8B-Instruct, and Llama-3-70B-Instruct-Gradient-1048k with PG-19-mini dataset. Across all models, TidalDecode consistently outperforms Quest, showing that TidalDecode with only two token selection layers can effectively retain critical information. TidalDecode achieves full accuracy with 64, 64, and 128 tokens in 10K-, 32K-, and 100K-context-length tests, which is only 0.6%, 0.2%, and 0.1% of total input lengths, respectively.

| Model (context length) | Method / Budget | K=32 | K=64 | K=128 | K=256 | K=512 |
|---|---|---|---|---|---|---|
| Llama-3-8B (10K) | Quest | 74% | 84% | 99% | 98% | **100%** |
| | TD+L13(Ours) | 88% | **98%** | **100%** | **100%** | **100%** |
| | TD+L15(Ours) | **92%** | 88% | 94% | 94% | **100%** |
| Llama-3-8B (100K) | Quest | 38% | 50% | 65% | 87% | 98% |
| | TD+L13(Ours) | **86%** | **92%** | **100%** | **100%** | **100%** |
| | TD+L15(Ours) | 84% | 90% | 92% | 98% | **100%** |
| Llama-3.1-8B (10K) | Quest | 74% | 86% | 94% | **100%** | 98% |
| | TD+L13(Ours) | **100%** | **100%** | **100%** | **100%** | **100%** |
| | TD+L14(Ours) | 98% | **100%** | **100%** | **100%** | **100%** |
| Llama-3.1-8B (32K) | Quest | 78% | 88% | 92% | **100%** | **100%** |
| | TD+L13(Ours) | **98%** | **100%** | **100%** | **100%** | **100%** |
| | TD+L14(Ours) | 80% | 98% | **100%** | **100%** | **100%** |
| Llama-3-70B (10K) | Quest | 68% | 72% | 90% | 98% | **100%** |
| | TD+L14(Ours) | 87% | 93% | **100%** | **100%** | **100%** |
| | TD+L31(Ours) | **90%** | **97%** | **100%** | **100%** | **100%** |
| Llama-3-70B (32K) | Quest | 50% | 80% | 88% | 92% | 78% |
| | TD+L14(Ours) | **82%** | **98%** | **98%** | **100%** | **100%** |
| | TD+L31(Ours) | 80% | 82% | 92% | 98% | **100%** |

may discard essential tokens, while selection-based approaches (Quest) often fail to consistently identify the ground-truth tokens with the highest attention scores in long contexts. Since Quest is the current state-of-the-art approach on this task that leverages an estimated attention score on the page level to identify important tokens for sparse attention, we first run TidalDecode on the same test as Quest on the LongChat-7b-v1.5-32k model and obtained Table 1 with competitive performance. To demonstrate the effectiveness of TidalDecode on long-dependency tasks, we further evaluate TidalDecode on tasks with 10K-, 32K-, and 100K-context-window lengths with the Llama-3-70B, Llama-3-8B, Llama-3.1-8B model using the PG-19-mini dataset, shown in Table 2. To ensure fairness, both TidalDecode and Quest use dense attention in the first two layers. In each test, we inserted a random password within the text and tested whether the specific method could retrieve the password correctly.

From Table 2, TidalDecode consistently outperforms Quest and achieves full accuracy with an extremely high sparsity (about 99.5% across all context lengths and models). These results demonstrate TidalDecode can achieve state-of-the-art performance with only two token selection layers. While Quest relies on page-level importance estimation for token selection, TidalDecode's exact selection with token reuse approach proves more effective for this task. Also, note that TidalDecode can reduce the token budget by up to $8\times$ when achieving a 100% accuracy compared with Quest. This further demonstrates that TidalDecode's exact token selection layer can obtain more relevant tokens than Quest.

### 4.2.2 LANGUAGE MODELING

Perplexity measures the negative likelihood of how well a model predicts the next word in a sequence, with lower values indicating better performance. We evaluate TidalDecode on Llama-3-8B-Instruct-Gradient-1048k with the PG-19 dataset, which includes up to 100 books, providing a comprehensive long-context benchmark.

As shown in Figure 4, TidalDecode+L9/13/15 consistently achieves lower perplexity than Quest across all token budget options (2048, 4096). This indicates that TidalDecode's position persistent sparse attention mechanism can effectively retain critical information without significantly sacrificing

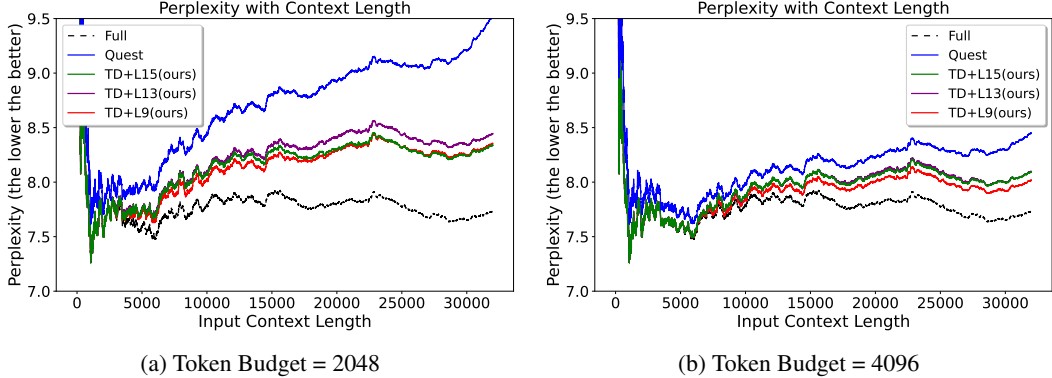

(a) Token Budget = 2048                                (b) Token Budget = 4096

Figure 4: Perplexity evaluation on the PG-19 dataset from 0 to 32K tokens. The results compare TidalDecode with different token re-selection layers (L9, L13, L15) to Quest across token budgets (2048 4a, 4096 4b). Lower perplexity indicates better model performance. Full refers to dense attention as baseline.

model accuracy, even as the sequence length grows, demonstrating its robustness for long-context inputs.

### 4.2.3 LONGBENCH EXPERIMENT

Table 3: Performance comparison on eight LongBench datasets evaluating single/multi-document QA, summarization, and retrieval tasks using Llama-3-8B-Instruct-Gradient-1048k. TidalDecode outperforms Quest at a 4096 token budget and achieves an average score higher than full-weight attention. The maximum F1-score for each task is in bold.

| Method (K)/Task | | MFQA | NrtQA | Qasp | 2Wiki | HotQA | QMSm | TrQA | PRe | Avg |
|---|---|---|---|---|---|---|---|---|---|---|
| Full | | 30.76 | 5.52 | **14.56** | 13.32 | 11.50 | 19.43 | **86.56** | 77.00 | 32.33 |
| Quest | (1024) | 26.21 | 4.08 | 12.19 | 12.61 | 10.75 | 19.56 | 83.47 | 63.84 | 29.09 |
| TD+L13 | (1024) | 28.57 | **7.63** | 11.11 | 13.56 | 9.82 | **20.37** | 79.78 | 75.17 | 30.75 |
| Quest | (4096) | 28.92 | 3.74 | 13.63 | 12.83 | 12.15 | 19.36 | 85.91 | 72.50 | 31.13 |
| TD+L13 | (4096) | **30.94** | 6.19 | 13.85 | **14.40** | **13.71** | 19.48 | 86.30 | **78.00** | **32.86** |

We also evaluate TidalDecode on LongBench, a benchmark designed to test LLMs on long-context tasks across diverse NLP domains (Bai et al., 2023). We focus on eight tasks: MultiFieldQA (MFQA) (Bai et al., 2023), NarrativeQA (NrtQA) (Kočiský et al., 2018), Qasper (Qasp) (Dasigi et al., 2021), 2WikiMQA (2Wiki) (Ho et al., 2020), HotpotQA (HotQA) (Yang et al., 2018), QMSum (QMSm) (Zhong et al., 2021), TriviaQA (TrQA) (Joshi et al., 2017), and Passage Retrieval (PRe) (Bai et al., 2023), which collectively composite a comprehensive evaluation benchmark in single/multi-document QA, summarization, and retrieval.

We evaluate all methods with Llama-3-8B-Instruct-Gradient-1048k. TidalDecode is compared against full-weight attention and Quest at token budgets of 1024 and 4096. As shown in Table 3, TidalDecode consistently outperforms Quest on all tasks at $K = 4096$ and on five tasks at $K = 1024$. Surprisingly, TidalDecode, in most cases, matches or exceeds full attention baseline with notable utilization rate on input tokens: 14% on NrtQA, 50% on MFQA, 80% on Qasp, 50% on 2WikiMQA, 32% on HotQA, 29% on QMSm, 35% on TrQA, and 33% on PRe. We hypothesize this is because TidalDecode's token selection process can filter out irrelevant information, thus leading to higher performance. We further include the results with cache correction in Appendix A.2.

These results demonstrate TidalDecode's generic ability to select tokens with the highest attention scores, achieving competitive or superior performance while significantly reducing token usage, making it ideal for long-context scenarios.

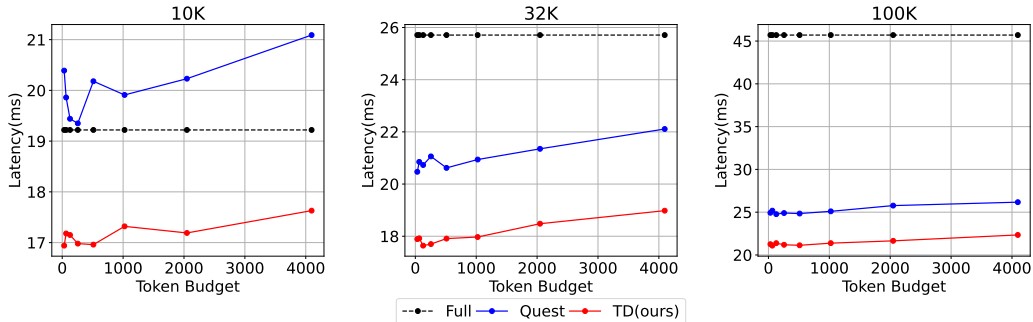

Figure 5: End-to-end latency results on Llama-2-7B model for Full attention baseline(Full), Quest, and TidalDecode(TD) when context length is 10K, 32K, and 100K, respectively.

### 4.3 EFFICIENCY EVALUATION

To show the efficiency of TidalDecode, we write customized kernels for our approach and measure the end-to-end decoding latency. We conduct evaluation under the configuration of Llama-2-7B on one Nvidia A100 (80 GB HBM, SXM4) with CUDA 12.2. We compare TidalDecode with state-of-the-art full attention serving library FlashInfer (Ye et al., 2024) and also the Quest implementation. As shown in Figure 5, we can observe that TidalDecode can consistently outperform full attention baseline and Quest by a large margin under all token budgets and context lengths. TidalDecode achieves this through token pattern reuse to minimize the token selection overhead. Notice that the latest Llama-3 model shares the same architecture as Llama-2, except it uses Group-Query-Attention instead of Multi-Head-Attention. However, this does not affect the relative efficiency comparison against Quest and full attention.

In Figure 6, we compare the overall attention latency between different methods on the Llama model with 32/64 layers. For the 32-layer Llama model, we have 2 full attention layers + 2 token selection layers + 28 sparse attention layers, while Quest has 2 full attention layers + 30 Quest attention layers. For the 64-layer Llama model, we have 2 full attention layers + 2 token selection layers + 60 sparse attention layers, while Quest has 2 full attention layers + 62 Quest attention layers. Thus, by completely removing the token estimation overhead in the sparse attention layers, for the 32-layer and 64-layer Llama model under all context lengths, TidalDecode can consistently achieve the lowest serving latency while bringing up to $5.56\times$ speed-up ratio against the full attention baseline and $2.17\times$ speed-up ratio against Quest. When the context length is 10K, Quest has a higher latency due to the token selection overhead, which aligns with the end-to-end results in Figure 5. In contrast, TidalDecode still achieves significant speed-up by utilizing the position persistent sparse attention mechanism. For the kernel-level analysis, please refer to Appendix A.5 for more details.

### 4.4 SENSITIVITY ANALYSIS ON TOKEN RE-SELECTION LAYER

In this section, we conduct sensitivity studies for different choices of the token re-selection layer. As TidalDecode only has one token re-selection layer in the middle, it is critical to choose the best-performed one. As shown in Figure 7, we have two interesting findings: (1). Different choices of token re-selection layers can significantly affect the accuracy of the results (2). For models within the same model family, the optimal token re-selection layer is consistent over different tasks. In our setup, the optimal token re-selection layer for the Llama-2-7B model is layer 7, while for the Llama-3-8B/Llama-3.1-8B model is layer 13. A concurrent KV cache compression work also identifies that layer 13 is surprisingly important for their approach as well (Shi et al., 2024). For a more detailed sensitivity results on the choice of different token re-selection layers, please refer to the appendix for more results.

## 5 CONCLUSION

To conclude, we introduce TidalDecode, an efficient LLM decoding framework with sparse attention. On observing the correlation of the pattern of tokens with the highest attention scores across different consecutive layers, TidalDecode proposes only to select tokens twice: once at the beginning layers and once in the middle layer to serve as a token re-selection layer. We find that using two token selection layers is necessary and sufficient to achieve high-generation quality. Additionally, by

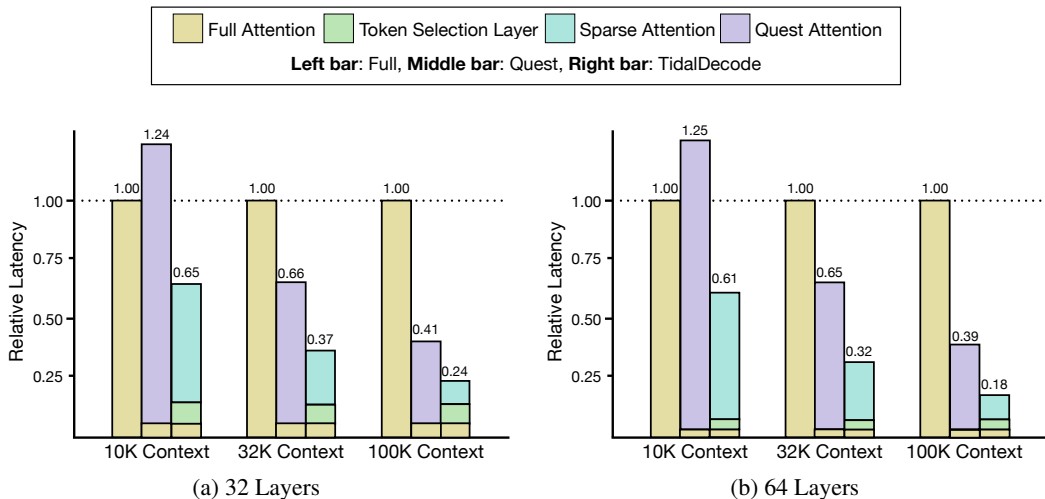

Figure 6: Overall attention latency results for different methods on the LLaMA model with (a) 32 and (b) 64 layers. We use the full attention model as a reference and show TidalDecode and Quest's overall attention latency ratio. For each group of the bar plots, the left/middle/right bar denotes the full attention baseline, Quest, and TidalDecode, respectively.

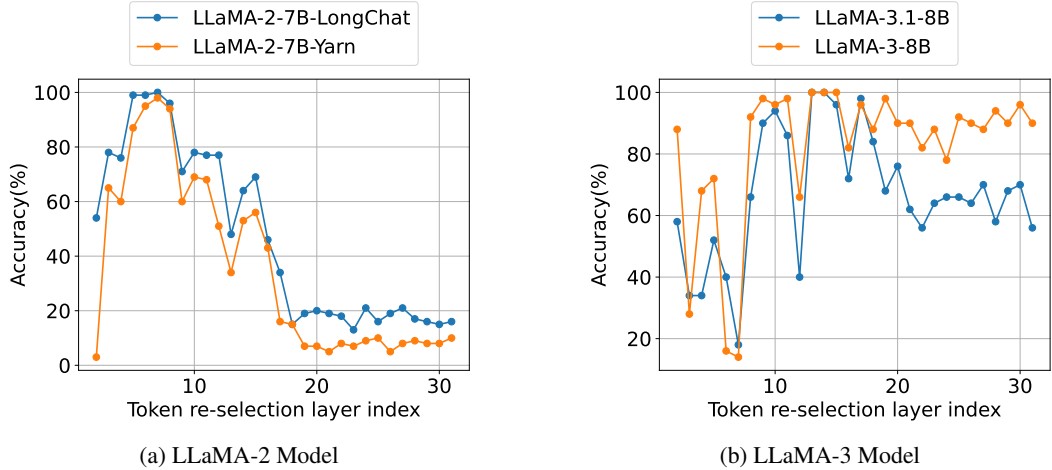

Figure 7: Sensitivity study on the choice of different token re-selection layer. We evaluate LLaMA-2-7B-LongChat, LLaMA-2-7B-Yarn, LLaMA-3-8B, and LLaMA-3.1-8B with TidalDecode with a token budget of 256.

reusing the token patterns throughout the sparse attention layer, TidalDecode greatly reduces the token selection overhead, resulting in a significant end-to-end speed-up ratio against existing methods. More interestingly, the optimal choice of the token re-selection layer is consistent across different tasks if the model is in the same model family.

## ACKNOWLEDGMENTS

This research is supported by NSF awards CNS-2147909, CNS-2211882, and CNS-2239351, and research awards from Amazon, Cisco, Google, Meta, NVIDIA, Oracle, Qualcomm, and Samsung. The views and conclusions contained in this document are those of the authors and should not be interpreted as representing the official policies, either expressed or implied, of any sponsoring institution, the U.S. government or any other entity.

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

# A APPENDIX

## A.1 EXTENDED LONGBENCH FOR LLAMA-3-8B-INSTRUCT

Table 4: Generation length for all tasks on LongBench evaluation.

| Method (K)/Task | Single-Doc QA | | | Multi-Doc QA | | Summarization | | Few-shot | Retrieval | Code Completion | | |
|---|---|---|---|---|---|---|---|---|---|---|---|---|
| | MFQA | NrtvQA | Qasp | 2Wiki | HotQA | GovRep | QSum | TrivQA | PRe | RB | LCC | Avg |
| Gen Length (Tokens) | 64 | 128 | 128 | 32 | 32 | 512 | 512 | 32 | 32 | 64 | 64 | 145 |

Table 5: Performance comparison on eleven LongBench datasets evaluating single/multi-document QA, summarization, retrieval tasks, and coding completion using Llama-3-8B-Instruct-Gradient-1048k. TidalDecode outperforms Quest at a 4096 token budget and achieves an average score higher than full-weight attention. The maximum F1-score for each task is in bold. Compared to Table 3, this table includes more results from summarization task (GovReport), coding tasks (RepoBench (RB) and LCC), and updated overall average scores.

| Method (K)/Task | | Single-Doc QA | | | Multi-Doc QA | | Summarization | | Few-shot | Retrieval | Code Completion | | |
|---|---|---|---|---|---|---|---|---|---|---|---|---|---|
| | | MFQA | NrtvQA | Qasp | Wiki | HotQA | GovRep | QSum | TrivQA | PRe | RB | LCC | Avg |
| Full | | 30.76 | 5.52 | **14.56** | 13.32 | 11.50 | 16.77 | 19.43 | **86.56** | 77.00 | 43.97 | 44.88 | 33.11 |
| Quest | (1024) | 26.21 | 4.08 | 12.19 | 12.61 | 10.75 | 15.89 | 19.56 | 83.47 | 63.84 | **49.43** | **49.66** | 31.61 |
| TD+L13 | (1024) | 28.57 | **7.63** | 11.11 | 13.56 | 9.82 | 16.41 | **20.37** | 79.78 | 75.17 | 43.29 | 40.67 | 31.49 |
| Quest | (4096) | 28.92 | 3.74 | 13.63 | 12.83 | 12.15 | 16.71 | 19.36 | 85.91 | 72.50 | 43.87 | 43.84 | 32.13 |
| TD+L13 | (4096) | 30.94 | 6.19 | 13.85 | **14.40** | **13.71** | **16.78** | 19.48 | 86.30 | **78.00** | 44.54 | 44.27 | **33.50** |

In Table 5, TidalDecode still achieves the highest score on added summarization task GovReport, where the model generates 512 tokens for each question. For code completion tasks, when selecting top-4096 tokens, TidalDecode outperforms full attention and Quest on RepoBench and stays closely to full attention on LCC. Notably, TidalDecode still achieves the highest average score across all eleven tasks.

## A.2 LONGBENCH FOR LLAMA-3-8B-INSTRUCT WITH CACHE CORRECTION

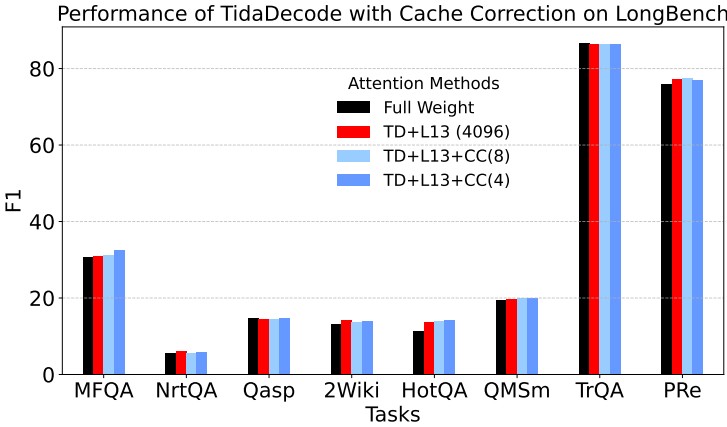

Figure 8: Cache correction (CC) is applied to TidalDecode for every 4/8 decoding steps following the description in Section 3.2, where we perform a forward pass with full attention on generated tokens from the last correction to correct the polluted tokens in the KV cache. We measured the performance of TidalDecode with cache correction on the same eight tasks as in Table 3 and compare that to full attention and basic TidalDecode with the same reselection layer (L13) and token budget (4096).

As illustrated in Figure 8, incorporating cache correction enhances the performance of the baseline TidalDecode (TD+L13+4096) across the same tasks in Table 3. Specifically, cache correction with a stride of 4 improves performance in five out of eight tasks, while a stride of 8 yields improvements

in six out of eight tasks. These results highlight the efficacy of cache correction in mitigating the adverse effects of polluted KV representations, which stem from sparse attention mechanisms and can accumulate errors, ultimately degrading model performance. Furthermore, as shown in Table 4, the tasks evaluated exhibit a wide range of generated sequence lengths, from 32 tokens in the single-document QA task HotQA to 512 tokens in the summarization task QMSm. The consistent improvements observed across this spectrum of sequence lengths underscore the robustness and generalizability of cache correction in various tasks.

## A.3  LONGBENCH FOR LLAMA-3.1-8B-INSTRUCT

Table 6: Performance comparison on eight LongBench datasets evaluating single/multi-document QA, summarization, and retrieval tasks using Llama-3.1-8B-Instruct. The maximum F1-score for each task is in bold.

| Method | MFQA | NrtQA | Qasp | 2Wiki | HotQA | QSm | TrQA | Pre | Avg |
|--------|------|-------|------|-------|-------|-----|------|-----|-----|
| Full | **27.02** | 25.59 | **13.05** | 16.64 | **16.86** | **23.88** | 91.48 | **97.67** | **39.02** |
| Quest (1024) | 22.35 | 14.89 | 12.44 | 14.24 | 14.12 | 23.86 | 81.71 | 95.73 | 34.92 |
| TD+13 (1024) | 23.70 | 23.25 | 11.14 | 13.53 | 13.72 | 22.69 | **92.35** | 92.15 | 36.57 |
| Quest (4096) | 26.34 | 21.17 | 11.99 | 15.61 | 16.26 | 23.61 | 90.73 | 96.35 | 37.76 |
| TD+13 (4096) | 25.89 | **26.29** | 12.65 | **16.86** | 15.94 | 23.27 | 90.22 | 95.47 | 38.32 |

TidalDecode and full-weight attention share the maximum F1 scores for all tasks, achieving the best scores in three tasks (NrtQA, 2Wiki, and TrQA). TidalDecode significantly outperforms Quest in 4/8 tasks (NrtQA, Qasp, 2Wiki, and TrQA) and full-attention in 3/8 tasks (NrtQA, 2Wiki, and TrQA). For other tasks, we stay close to the full attention and also obtains a higher average score than Quest.

## A.4  END-TO-END EFFICIENCY EVALUATION RESULTS

Table 7: TidalDecode end-to-end efficiency results on LLaMA-2-7B

| Context Length | Full Attention (ms) | TidalDecode(ms) | | | | | | | |
|----------------|---------------------|------|------|-------|-------|-------|--------|--------|--------|
| | | K=32 | K=64 | K=128 | K=256 | K=512 | K=1024 | K=2048 | K=4096 |
| 10K | 19.22 | 16.94 | 17.18 | 17.15 | 16.98 | 16.96 | 17.32 | 17.19 | 17.63 |
| 32K | 25.71 | 17.89 | 17.92 | 17.64 | 17.70 | 17.91 | 17.97 | 18.48 | 18.98 |
| 100K | 45.70 | 21.26 | 21.09 | 21.38 | 21.19 | 21.13 | 21.38 | 21.65 | 22.34 |

Table 8: Quest end-to-end efficiency results on LLaMA-2-7B

| Context Length | Full Attention (ms) | TidalDecode(ms) | | | | | | | |
|----------------|---------------------|------|------|-------|-------|-------|--------|--------|--------|
| | | K=32 | K=64 | K=128 | K=256 | K=512 | K=1024 | K=2048 | K=4096 |
| 10K | 19.22 | 20.39 | 19.86 | 19.44 | 19.35 | 20.18 | 19.91 | 20.23 | 21.09 |
| 32K | 25.71 | 20.47 | 20.85 | 20.73 | 21.06 | 20.62 | 20.94 | 21.35 | 22.11 |
| 100K | 45.70 | 24.93 | 25.18 | 24.77 | 24.90 | 24.84 | 25.10 | 25.77 | 26.17 |

## A.5 KERNEL LATENCY

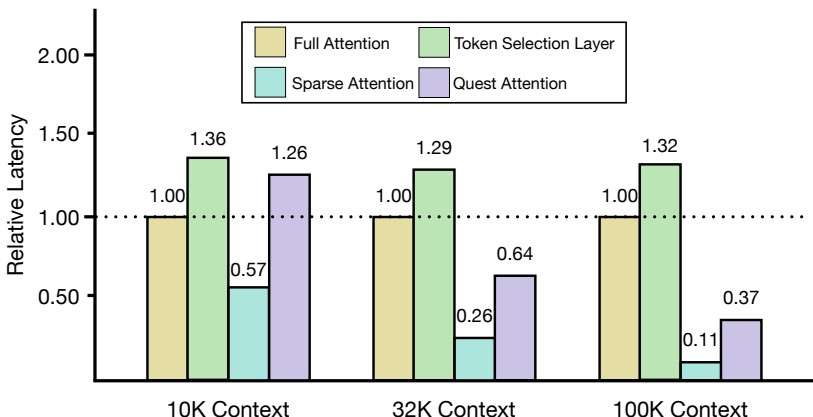

Figure 9: The breakdown latency results for the full attention, token selection attention, sparse attention, and Quest attention kernels over 10K, 32K, and 100K context length. We use full attention latency as a reference and report other kernels' relative latency ratio. We use a token budget of K=512 for TidalDecode and Quest across all evaluations.

In Figure 9, we further break down the latency comparison for different attention modules to show why TidalDecode can bring significant speed-up consistently. We compare different attention modules, namely, the full attention layer, the token selection layer, TidalDecode's sparse attention layer, and the Quest attention layer over the 10K, 32K, and 100K context length. We can observe that, as TidalDecode's sparse attention kernel can directly reuse previous token patterns, it completely removes the important token estimation overhead in the Quest attention kernel, resulting in up to 3.36× speed-up compared with the Quest implementation. On the other hand, even though TidalDecode's token selection layer has a slightly higher latency, we only have two token selection layers even in the 70B LLaMA model that has 64 layers in total.

## A.6 FULL SENSITIVITY STUDIES ON DIFFERENT TOKEN RE-SELECTION LAYER

Table 9: Sensitivity study of re-selection layer (RL) on 10k-context-length Needle-in-the-Haystack test for LLaMA-3.2-3B-Instruct with TidalDecode. The best accuracy for each token budget (K) is in bold.

| RL/K | 32 | 64 | 128 | 256 | 512 |
|---|---|---|---|---|---|
| TidalDecode+L2 | 2% | 12% | 14% | 16% | 30% |
| TidalDecode+L3 | 2% | 6% | 8% | 10% | 24% |
| TidalDecode+L4 | 6% | 14% | 16% | 20% | 28% |
| TidalDecode+L5 | 2% | 10% | 22% | 26% | 36% |
| TidalDecode+L6 | 18% | 26% | 26% | 32% | 46% |
| TidalDecode+L7 | 6% | 10% | 16% | 18% | 28% |
| TidalDecode+L8 | 20% | 20% | 46% | 60% | 84% |
| TidalDecode+L9 | 6% | 12% | 32% | 58% | 66% |
| TidalDecode+L10 | 44% | 58% | 50% | 60% | 64% |
| TidalDecode+L11 | 4% | 12% | 16% | 22% | 28% |
| TidalDecode+L12 | **50%** | **84%** | **96%** | **98%** | 98% |
| TidalDecode+L13 | 42% | 80% | 94% | **98%** | **100%** |
| TidalDecode+L14 | 28% | 44% | 54% | 60% | 72% |
| TidalDecode+L15 | 2% | 8% | 16% | 22% | 36% |
| TidalDecode+L16 | 4% | 16% | 12% | 22% | 34% |
| TidalDecode+L17 | 2% | 6% | 16% | 20% | 32% |
| TidalDecode+L18 | 2% | 10% | 12% | 18% | 28% |
| TidalDecode+L19 | 2% | 6% | 10% | 18% | 32% |
| TidalDecode+L20 | 6% | 10% | 12% | 18% | 24% |
| TidalDecode+L21 | 6% | 8% | 10% | 16% | 26% |
| TidalDecode+L22 | 6% | 0% | 12% | 12% | 26% |
| TidalDecode+L23 | 4% | 14% | 14% | 18% | 26% |
| TidalDecode+L24 | 2% | 10% | 16% | 20% | 28% |
| TidalDecode+L25 | 4% | 8% | 14% | 16% | 22% |
| TidalDecode+L26 | 2% | 10% | 10% | 22% | 26% |
| TidalDecode+L27 | 0% | 10% | 12% | 22% | 26% |
| Quest | 46% | 56% | 72% | 88% | 96% |

Table 10: Sensitivity study of re-selection layer (RL) on 10k-context-length Needle-in-the-Haystack test for LLaMA-3.1-8B-Instruct with TidalDecode. The best accuracy for each token budget (K) is in bold. Layer 13 and Layer 14 are the best two re-selection layers for accuracy.

| RL/K | 32 | 64 | 128 | 256 |
|---|---|---|---|---|
| TidalDecode+L2 | 36% | 38% | 46% | 58% |
| TidalDecode+L3 | 8% | 10% | 14% | 34% |
| TidalDecode+L4 | 0% | 10% | 16% | 34% |
| TidalDecode+L5 | 14% | 30% | 52% | 52% |
| TidalDecode+L6 | 8% | 12% | 28% | 40% |
| TidalDecode+L7 | 6% | 10% | 10% | 18% |
| TidalDecode+L8 | 34% | 44% | 50% | 66% |
| TidalDecode+L9 | 64% | 78% | 82% | 90% |
| TidalDecode+L10 | 56% | 74% | 84% | 94% |
| TidalDecode+L11 | 52% | 76% | 82% | 86% |
| TidalDecode+L12 | 8% | 10% | 28% | 40% |
| TidalDecode+L13 | **100%** | **100%** | **100%** | **100%** |
| TidalDecode+L14 | 98% | **100%** | **100%** | **100%** |
| TidalDecode+L15 | 56% | 78% | 88% | 96% |
| TidalDecode+L16 | 18% | 46% | 54% | 72% |
| TidalDecode+L17 | 64% | 74% | 86% | 98% |
| TidalDecode+L18 | 64% | 70% | 74% | 84% |
| TidalDecode+L19 | 58% | 50% | 60% | 68% |
| TidalDecode+L20 | 68% | 60% | 62% | 76% |
| TidalDecode+L21 | 40% | 48% | 48% | 62% |
| TidalDecode+L22 | 28% | 38% | 46% | 56% |
| TidalDecode+L23 | 40% | 46% | 52% | 64% |
| TidalDecode+L24 | 30% | 46% | 54% | 66% |
| TidalDecode+L25 | 40% | 54% | 50% | 66% |
| TidalDecode+L26 | 34% | 48% | 62% | 64% |
| TidalDecode+L27 | 38% | 50% | 54% | 70% |
| TidalDecode+L28 | 30% | 40% | 56% | 58% |
| TidalDecode+L29 | 32% | 48% | 56% | 68% |
| TidalDecode+L30 | 36% | 48% | 52% | 70% |
| TidalDecode+L31 | 30% | 36% | 42% | 56% |

Table 11: Sensitivity study of re-selection layer (RL) on 10k-context-length Needle-in-the-Haystack test for LLaMA-3-70B-Instruct-Gradient-1048k; we first run top_k = 512 and filter out those layers that do not achieve full accuracy with TidalDecode. The best accuracy for each token budget (K) is in bold. Layer 14 and Layer 31 are the best two Re-selection layers for accuracy.

| RL/K | 32 | 64 | 128 | 256 | 512 | RL/K | 32 | 64 | 128 | 256 | 512 |
|---|---|---|---|---|---|---|---|---|---|---|---|
| L2 | - | - | - | - | 6% | L33 | 50% | 87% | 90% | 93% | **100%** |
| L3 | - | - | - | - | 37% | L34 | - | - | - | - | 97% |
| L4 | - | - | - | - | 23% | L35 | 70% | 83% | **100%** | **100%** | **100%** |
| L5 | - | - | - | - | 63% | L36 | 50% | 83% | 97% | 97% | **100%** |
| L6 | - | - | - | - | 70% | L37 | - | - | - | - | 90% |
| L7 | - | - | - | - | 90% | L38 | 37% | 83% | 83% | 80% | **100%** |
| L8 | - | - | - | - | 70% | L39 | - | - | - | - | 87% |
| L9 | - | - | - | - | 30% | L40 | - | - | - | - | 50% |
| L10 | - | - | - | - | 83% | L41 | - | - | - | - | 97% |
| L11 | - | - | - | - | 70% | L42 | - | - | - | - | 53% |
| L12 | - | - | - | - | 63% | L43 | - | - | - | - | 67% |
| L13 | - | - | - | - | 50% | L44 | - | - | - | - | 83% |
| L14 | 87% | 93% | **100%** | **100%** | **100%** | L45 | - | - | - | - | 70% |
| L15 | - | - | - | - | 97% | L46 | - | - | - | - | 63% |
| L16 | - | - | - | - | 63% | L47 | - | - | - | - | 77% |
| L17 | - | - | - | - | 87% | L48 | - | - | - | - | 97% |
| L18 | 50% | 70% | 83% | 97% | **100%** | L49 | - | - | - | - | 77% |
| L19 | - | - | - | - | 93% | L50 | - | - | - | - | 70% |
| L20 | - | - | - | - | 87% | L51 | - | - | - | - | 93% |
| L21 | 53% | 80% | 93% | 97% | **100%** | L52 | - | - | - | - | 77% |
| L22 | - | - | - | - | 97% | L53 | - | - | - | - | 70% |
| L23 | 53% | 93% | 97% | **100%** | **100%** | L54 | - | - | - | - | 60% |
| L24 | 33% | 60% | 77% | 93% | **100%** | L55 | - | - | - | - | 53% |
| L25 | - | - | - | - | 80% | L56 | - | - | - | - | 87% |
| L26 | - | - | - | - | 87% | L57 | - | - | - | - | 57% |
| L27 | 50% | 87% | 93% | 93% | **100%** | L58 | - | - | - | - | 50% |
| L28 | 80% | 83% | 93% | 87% | **100%** | L59 | - | - | - | - | 50% |
| L29 | - | - | - | - | 97% | L60 | - | - | - | - | 57% |
| L30 | 33% | 67% | 80% | 90% | **100%** | L61 | - | - | - | - | 30% |
| L31 | **90%** | **97%** | **100%** | **100%** | **100%** | L62 | - | - | - | - | 43% |
| L32 | 27% | 73% | 80% | 97% | **100%** | L63 | - | - | - | - | 43% |

Table 12: Sensitivity study of re-selection layer (RL) on 10k-context-length Needle-in-the-Haystack test for Llama-3-8B-Instruct-Gradient-1048k with TidalDecode. The best accuracy for each token budget (K) is in bold. Layer 9, Layer 13, and Layer 14 are the best three re-selection layers for accuracy.

| RL/K | 16 | 32 | 64 | 128 | 256 | 512 |
|---|---|---|---|---|---|---|
| TidalDecode+L2 | 78% | 84% | 76% | 94% | 88% | 98% |
| TidalDecode+L3 | 0% | 6% | 10% | 16% | 28% | 64% |
| TidalDecode+L4 | 2% | 10% | 16% | 28% | 68% | 84% |
| TidalDecode+L5 | 10% | 12% | 32% | 52% | 72% | 80% |
| TidalDecode+L6 | 4% | 6% | 10% | 14% | 16% | 24% |
| TidalDecode+L7 | 2% | 10% | 10% | 10% | 14% | 28% |
| TidalDecode+L8 | 26% | 64% | 80% | 90% | 92% | 96% |
| TidalDecode+L9 | 52% | 90% | 96% | **100%** | 98% | **100%** |
| TidalDecode+L10 | 72% | 76% | 86% | 94% | 96% | **100%** |
| TidalDecode+L11 | 56% | 74% | 94% | **100%** | 98% | 98% |
| TidalDecode+L12 | 8% | 14% | 22% | 44% | 66% | 94% |
| TidalDecode+L13 | **92%** | **92%** | **96%** | **100%** | **100%** | **100%** |
| TidalDecode+L14 | 74% | 68% | 88% | 98% | **100%** | **100%** |
| TidalDecode+L15 | 74% | 94% | 92% | 88% | **100%** | **100%** |
| TidalDecode+L16 | 44% | 50% | 72% | 66% | 82% | 94% |
| TidalDecode+L17 | 42% | 60% | 74% | 82% | 96% | 96% |
| TidalDecode+L18 | 60% | 72% | 74% | 74% | 88% | 98% |
| TidalDecode+L19 | 58% | 74% | 82% | 84% | **98%** | 96% |
| TidalDecode+L20 | 64% | 74% | **96%** | 78% | 90% | 98% |
| TidalDecode+L21 | 10% | 38% | 60% | 66% | 90% | 94% |
| TidalDecode+L22 | 60% | 70% | 68% | 72% | 82% | 98% |
| TidalDecode+L23 | 58% | 78% | 70% | 86% | 88% | 98% |
| TidalDecode+L24 | 62% | 58% | 76% | 70% | 78% | 92% |
| TidalDecode+L25 | 66% | 86% | 84% | 82% | 92% | **100%** |
| TidalDecode+L26 | 54% | 64% | 66% | 80% | 90% | 94% |
| TidalDecode+L27 | 84% | 80% | 94% | 96% | 88% | **100%** |
| TidalDecode+L28 | 66% | 66% | 76% | 84% | **94%** | 94% |
| TidalDecode+L29 | 72% | 80% | 88% | 80% | 90% | 96% |
| TidalDecode+L30 | 80% | **90%** | 86% | 88% | 96% | **100%** |
| TidalDecode+L31 | 74% | 90% | 88% | 84% | 90% | 96% |

Table 13: Sensitivity study of re-selection layer (RL) on 3k-context-length Needle-in-the-Haystack test for LongChat-7b-v1.5-32k with TidalDecode. The best accuracy for each token budget (K) is in bold. Layer 7 serves the best re-selection layer for accuracy.

| RL/K | 32 | 64 | 128 | 256 |
|---|---|---|---|---|
| TidalDecode+L2 | 2% | 2% | 6% | 54% |
| TidalDecode+L3 | 10% | 52% | 67% | 78% |
| TidalDecode+L4 | 4% | 36% | 65% | 76% |
| TidalDecode+L5 | 17% | 87% | 94% | 99% |
| TidalDecode+L6 | 70% | 96% | 99% | 99% |
| TidalDecode+L7 | **80%** | **98%** | **100%** | **100%** |
| TidalDecode+L8 | 58% | 82% | 96% | 96% |
| TidalDecode+L9 | 7% | 31% | 59% | 71% |
| TidalDecode+L10 | 16% | 59% | 71% | 78% |
| TidalDecode+L11 | 34% | 61% | 68% | 77% |
| TidalDecode+L12 | 17% | 32% | 53% | 77% |
| TidalDecode+L13 | 5% | 10% | 28% | 48% |
| TidalDecode+L14 | 24% | 41% | 57% | 64% |
| TidalDecode+L15 | 37% | 47% | 62% | 69% |
| TidalDecode+L16 | 16% | 24% | 28% | 46% |
| TidalDecode+L17 | 4% | 4% | 10% | 34% |
| TidalDecode+L18 | 2% | 3% | 8% | 15% |
| TidalDecode+L19 | 0% | 1% | 7% | 19% |
| TidalDecode+L20 | 0% | 3% | 6% | 20% |
| TidalDecode+L21 | 0% | 2% | 10% | 19% |
| TidalDecode+L22 | 0% | 4% | 4% | 18% |
| TidalDecode+L23 | 0% | 2% | 5% | 13% |
| TidalDecode+L24 | 0% | 2% | 6% | 21% |
| TidalDecode+L25 | 0% | 2% | 7% | 16% |
| TidalDecode+L26 | 0% | 1% | 7% | 19% |
| TidalDecode+L27 | 0% | 2% | 4% | 21% |
| TidalDecode+L28 | 1% | 2% | 10% | 17% |
| TidalDecode+L29 | 0% | 3% | 7% | 16% |
| TidalDecode+L30 | 1% | 1% | 9% | 15% |
| TidalDecode+L31 | 1% | 2% | 5% | 16% |

Table 14: Sensitivity study of re-selection layer (RL) on 3k-context-length Needle-in-the-Haystack test for Yarn-Llama-2-7b-128k with TidalDecode. The best accuracy for each token budget (K) is in bold. Layer 7 serves the best re-selection layer for accuracy.

| RL/K | 32 | 64 | 128 | 256 | 512 |
|---|---|---|---|---|---|
| TidalDecode+L2 | 0% | 0% | 0% | 3% | 25% |
| TidalDecode+L3 | 11% | 26% | 39% | 65% | 85% |
| TidalDecode+L4 | 5% | 17% | 34% | 60% | 92% |
| TidalDecode+L5 | 16% | 42% | 65% | 87% | 96% |
| TidalDecode+L6 | 73% | 83% | 89% | 95% | **100%** |
| TidalDecode+L7 | 73% | **95%** | **98%** | **98%** | **100%** |
| TidalDecode+L8 | **87%** | 92% | 97% | 94% | 99% |
| TidalDecode+L9 | 7% | 21% | 43% | 60% | 95% |
| TidalDecode+L10 | 12% | 31% | 58% | 69% | 93% |
| TidalDecode+L11 | 20% | 21% | 46% | 68% | 97% |
| TidalDecode+L12 | 2% | 15% | 28% | 51% | 92% |
| TidalDecode+L13 | 4% | 5% | 20% | 34% | 88% |
| TidalDecode+L14 | 16% | 20% | 49% | 53% | 91% |
| TidalDecode+L15 | 2% | 25% | 44% | 56% | 90% |
| TidalDecode+L16 | 10% | 13% | 21% | 43% | 86% |
| TidalDecode+L17 | 3% | 4% | 9% | 16% | 85% |
| TidalDecode+L18 | 0% | 1% | 2% | 15% | 84% |
| TidalDecode+L19 | 0% | 2% | 3% | 7% | 80% |
| TidalDecode+L20 | 0% | 2% | 0% | 7% | 79% |
| TidalDecode+L21 | 0% | 0% | 1% | 5% | 77% |
| TidalDecode+L22 | 0% | 0% | 0% | 8% | 76% |
| TidalDecode+L23 | 0% | 0% | 1% | 7% | 74% |
| TidalDecode+L24 | 0% | 2% | 0% | 9% | 73% |
| TidalDecode+L25 | 0% | 0% | 2% | 10% | 71% |
| TidalDecode+L26 | 0% | 1% | 1% | 5% | 70% |
| TidalDecode+L27 | 0% | 1% | 3% | 8% | 68% |
| TidalDecode+L28 | 0% | 1% | 0% | 9% | 67% |
| TidalDecode+L29 | 0% | 0% | 2% | 8% | 65% |
| TidalDecode+L30 | 0% | 1% | 1% | 8% | 64% |
| TidalDecode+L31 | 0% | 0% | 2% | 10% | 62% |

