# OpenReview forum: "TidalDecode: Fast and Accurate LLM Decoding with Position Persistent Sparse Attention"
_ICLR.cc/2025/Conference — ICLR 2025 Poster_

### Official Review · Reviewer_mweX · 2024-10-31

**Soundness:** 3
**Presentation:** 3
**Contribution:** 2
**Rating:** 6
**Confidence:** 4

**Summary:**

The paper presents TidalDecode to enhance the efficiency and accuracy of large language models (LLMs) through position persistent sparse attention. TidalDecode introduces token selection layers to identify and retain the most relevant tokens, with other layers using sparse attention to manage memory and reduce computational demands. Empirical evaluations indicate that TidalDecode can achieve up to a 2.1× reduction in latency while maintaining comparable generative performance to full attention mechanisms.

**Strengths:**

The proposed method is effective, maintaining the advantages of both eviction and selection-based sparse attention methods according to experiments.
The paper presents robust evaluations across various LLMs and tasks, including LongBench and Needle-in-the-Haystack tests. Results indicate that TidalDecode consistently performs well compared to state-of-the-art methods.

**Weaknesses:**

1. Definition of `spatial coherence`. The spatial coherence seems just come from nowhere. I can mostly understand it refers to the consecutive layers have significant attention overlaps. But there is not a `scientific definition` in this paper.
2. Complexity of Implementation: TidalDecode's reliance on custom GPU kernels and specific layer configurations could limit its accessibility and adoption. Please provide pseudo code or GPU consumption data, it could make the results more clear.
3. The proposed method can be considered as a kind of hierarchical attention, more related baselines can be included. such as [1,2]
4. KV-cache correction seems to be crucial, no ablation study (or I just missed). Please provide the results without KV-cache correction.
5. training details, e.g, learning rate, training tokens, GPU hours, etc. It would help the reproduction.
6. Does `L` matter in different models? How to determine the optimal hyper-parameters. It would be appreciated if automatic prediction of `L` or a guideline for `L` selection across different model size, model architecture, etc.
7. There might be some conflicts in the packages used in latex.

[1] Chalkidis, Ilias, et al. "An exploration of hierarchical attention transformers for efficient long document classification." arXiv preprint arXiv:2210.05529 (2022).
[2] Yang, Zichao, et al. "Hierarchical attention networks for document classification." Proceedings of the 2016 conference of the North American chapter of the association for computational linguistics: human language technologies. 2016.

**Questions:**

Please see weaknesses

---

> ### Author Response · Authors · 2024-11-20
> **Respond to reviewer mweX**
>
> We thank the reviewer for their review and address their main concerns below.
>
> `W1. Spatial Coherence`
>
> We thank the reviewer for pointing out our use of spatial coherence and the need for a definition. We have modified accordingly in the fourth paragraph of the introduction section with a definition of spatial coherence that is highlighted in the underlines.
>
> `W2. Complexity of Implementation`
>
> For the GPU kernel implementation, we are building on top of the state-of-the-art attention kernel implementation library FlashInfer, which has been used in vLLM, SGLang, MLC-LLM, etc. Thus, we believe that our implementation can be easily integrated into popular LLM serving systems. In addition, our kernel implementation code has been anonymized and included in the supplementary materials for your reference.
>
> `W3. Hierarchical Attention Networks`
>
> We thank the reviewer for suggesting potential related works. We want to clarify that TidalDecode is a training-free method that can be directly applied to existing pretrained models like LLaMA-2 and LLaMA-3 during the inference phase, which is different from the hierarchical attention methods [1, 2] that requires pre-training.
>
> `W4. KV-cache correction`
>
> Even though we have introduced the mechanism of KV cache correction, we don't use the feature for all our current experiments to make it a fair comparison against other baseline models that haven’t applied cache correction. Additionally, for most of the tasks in our evaluations, the generation length is insufficient to see a degradation in model quality, so the model performance is still maintained even without KV cache correction. However, to show the effects of KV cache correction, we provide additional experimental results for the LongBench evaluation with KV cache correction on the original eight tasks and include results in Figure 9 in the revised paper. Albeit the improvement is not significant due to a short generation length, cache correction with a stride of 4 improves performance in five out of eight tasks, and a stride of 8 yields improvements in six out of eight tasks. These results highlight the efficacy of cache correction in mitigating the adverse effects of polluted KV representations, which stem from sparse attention mechanisms and can accumulate errors, ultimately degrading model performance.
>
> `W5. Training details`
>
> Please refer to the response to W3, as our method is a training-free method.
>
> `W6. How to choose the optimal token re-selection layer (L)`
>
> Our observations show that the optimal token re-selection layer choice is consistent across different task and decoding steps for models within the same model family. We hypothesize that this is an intrinsic property resulting from specific models' pretraining process. This phenomenon is an interesting observation but lacks theoretical analysis, which we leave as a future work. To sum up, we don’t have an automatic layer selection strategy at the current stage since the optimal choice is consistent within the same model family. We can thus directly identify the optimal choice from sample data only once and use it as a guideline for all later deployments. If the optimal layer choice within a model family does vary for different downstream tasks or even different decoding steps, there is certainly a need to design a flexible, adaptive layer selection strategy.
>
> `W7. Conflicts in the packages used in latex.`
>
> We have tried our best to resolve all the warnings in the compilation for the revised paper. Please let us know if the package conflicts still persist.
>
> **References:**
>
> [1] Chalkidis, Ilias, et al. "An exploration of hierarchical attention transformers for efficient long document classification." arXiv preprint arXiv:2210.05529 (2022).
>
> [2] Yang, Zichao, et al. "Hierarchical attention networks for document classification." Proceedings of the 2016 conference of the North American chapter of the association for computational linguistics: human language technologies. 2016.

---

> ### Comment · Reviewer_mweX · 2024-11-25
>
> Thanks for the responses.
> 1. I'm confused by the response to `W4`. It doesn't make any sense to me that the authors introduced some methods, but didn't use them in the experiments. TBH the augments cannot convince me. It's just a `useless` method that would make the paper more "scientific" to me if the authors express such arguments: `Even though we have introduced the mechanism of KV cache correction, we don't use the feature for all our current experiments to make it a fair comparison against other baseline models that haven’t applied cache correction.`
> 2. Response to W6. As a summarization, the authors cannot provide any selection strategy, this will greatly reduce the real world application of the proposed method, as we can only try every L for optimal performance.  Even on a small dev test, it would be very unreliable and time-consuming.
>
> As a result, the author's responses did not address my concerns, I would like to keep the score unchanged.

---

> > ### Author Response · Authors · 2024-11-25
> >
> > We thank the reviewer for the response and would like to try our best to address your concerns.
> >
> > 1. We agree with the reviewer that having ablation results with KV cache correction can make our evaluation more comprehensive. Therefore, we have included ablation studies on KV cache correction in Figure 9 in the appendix. We can see slight improvement over the version without KV cache correction. More importantly, from all other evaluations, we can observe that our core method (position persistent sparse attention) can consistently achieve the best performance while bringing a significant speed-up ratio due to its simplicity, even without KV cache correction.
> >
> > 2. We agree with the reviewer that an automatic selection method is required if the optimal choice of token re-selection layer is sensitive to different downstream tasks. However, our evaluation results demonstrate that the optimal token re-selection layer is really consistent across all downstream tasks and even all the models within the same model family. For instance, the optimal token re-selection layer is the 13th layer for LLaMA-3/3.1/3.2-8B in all downstream tasks we evaluated. Therefore, we only need to decide the optimal token re-selection layer once, and the selected layer can be used for all later deployments over different tasks. Additionally, as the model performance is really sensitive to different choices of the optimal token re-selection layer (as shown in Table 9-14), we can identify the optimal token re-selection layer with minimal effort through a needle-in-the-haystack task over a small set (100) of synthetic examples. Additionally, in Figures 2(a) and 2(b), we present our observation, which can serve as a method to effectively narrow down the search space for selecting reselection layers. This is achieved by calculating the average overlap ratio when different layers are chosen as the reselection layer in a simple Needle-in-the-Haystack setting. This approach further reduces the effort required to identify the optimal reselection layer.
> >
> > We hope our responses above could address most of your concerns, and please feel free to leave more questions if you have. We greatly appreciate your feedback, which deeply contributes to a better revision of our previous draft.

---

### Official Review · Reviewer_oUXc · 2024-11-01

**Soundness:** 3
**Presentation:** 3
**Contribution:** 3
**Rating:** 6
**Confidence:** 4

**Summary:**

This article introduces a new sparse attention mechanism, TidalDecode, which is an efficient selection-based attention computing mechanism for the second stage of LLM inference, specifically the decoding stage. The authors discovered through empirical experiments that there is a significant overlap among the Top-K tokens in the attention scores between consecutive Transformer layers. Thus, they propose performing full attention computation only in the initial and intermediate layers, while the other layers select the most relevant KV-cache to compute attention, achieving an efficient sparse attention mechanism. Experimental results across diverse LLMs and tasks demonstrate significant improvements in both decoding latency and the quality of generated results. The optimal choice of the token re-selection layer is consistent across different tasks if the model belongs to the same model family.

**Strengths:**

1.The article is written in a clear and comprehensible way, with strong consistency across empirical experiments, motivation, methodology, and the effectiveness of the proposed method.
2.The method is simple yet effective, supported by extensive experimental and theoretical validations. A wide range of diverse experiments, including multiple LLMs and tasks, demonstrates the excellent performance of the proposed method.

**Weaknesses:**

1.The chosen token across different tasks for the same model family is almost consistent; however, the dramatic variation in performance between L12 and L13 in Table 7 presents a significant challenge for the method's generalization across different models and datasets. This variability could lead to catastrophic results when generalizing to other datasets.
2.Regarding Figure 2a, the overlap results do not seem significant based solely on the visual representation. Please provide a detailed explanation of the information that can be interpreted from the figure.

**Questions:**

1.Building on Weakness 1, an adaptive layer selection strategy seems necessary. Did the authors explore any strategies in this regard?
2.I am quite curious about the actual application effects of KV Cache Correction. Were corresponding experimental validations conducted?
3.In Table 3, it is interesting to note that on certain datasets like HotQA, the sparse attention mechanism sometimes outperforms the full attention mechanism. Is there a relevant explanation for this, or an empirical rationale behind it?

---

> ### Author Response · Authors · 2024-11-20
> **Respond to reviewer oUXc**
>
> We thank the reviewer for their review and address their main concerns below.
>
> `Q1/W1. Adaptive layer selection strategy`
>
> We agree with the reviewer that the model's performance is highly sensitive to the choice of the token re-selection layer. However, as you have suggested, our observations show that the optimal token re-selection layer choice is consistent across different task and decoding steps for models within the same model family. We hypothesize that this is an intrinsic property resulting from specific models' pretraining process. This phenomenon is an interesting observation but lacks theoretical analysis, which we leave as a future work. To sum up, we don’t have an adaptive layer selection strategy at the current stage since the optimal choice is consistent within the same model family. We can thus directly identify the optimal choice from sample data only once and use it for all later deployments. If the optimal layer choice within a model family does vary for different downstream tasks or even different decoding steps, there is certainly a need to design a flexible, adaptive layer selection strategy.
>
> `W2. Figure 2`
>
> We thank the reviewer for pointing out the confusion in our presentation for Figure 2. We have updated our descriptions and definitions for terms (e.g., recall rate) in the text and the caption, which have been highlighted in blue. More specifically, Figure 2a shows the overlap ratio of the top-k tokens selected between different layers, which is used to calculate the recall rate in Figure 2b. The recall rate is defined as the averaged overlap ratio in Figure 2a over the sparse attention layers between the third layer and the corresponding token re-selection layer, and over all the sparse layers after the chosen re-selection layer. For instance, the recall rate for choosing layer 13 as the token re-selection layer is the average of the overlap ratio over the two red bars in Figure 2a. Then, we can clearly see that adding a token re-selection layer around layer 13 can significantly boost the recall rate due to the improvement in the overlap ratio from the purple bar under layer 3 to the red bar under layer 13 in Figure 2a. Figure 2b quantitatively captures the improvement. We can observe that selecting layer 13 as the token re-selection layer is optimal if we only allow one additional token re-selection layer.
>
> The reason why the overlap is not that significant is two-folded: 1. As the context length that we used is 100K and the token budget is 256, an overlap ratio of 40% is relatively significant compared to the sparsity ratio of 1/400. However, the heatmap we used for Figure 2a allows a maximum value of 1, making 0.4 less obvious. 2. Besides the token selection overlap ratio, we have added additional evidence that supports our claim in Figure 8 in the appendix. The figure captures the cosine similarity between adjacent layers’ attention scores, and we have provided our justifications along with the figure. The reason why we use cosine similarity is that the token overlap ratio only focuses on the number of shared tokens rather than the contribution of each token in terms of the attention score. For instance, two layers can have only a few tokens that are shared out of the 256 tokens we chose, but these several tokens can contribute a huge portion of the attention scores, which can be captured by the cosine similarity metrics we used in Figure 8. The resulting attention score similarity (correlation) for adjacent layers in Figure 8 is much higher than the token overlap ratio in Figure 2a, which provides further evidence for our design.
>
> `Q2. KV Cache Correction`
>
> For most of the tasks in our evaluations, the generation length is insufficient to see a degradation in model quality. On the other hand, to make it a fair comparison against other baseline models that haven’t applied cache correction, we don’t use the KV cache correction feature for all our current experiments. However, to provide some results on KV cache correction, we provide additional experimental results for the LongBench test with KV cache correction in Figure 9 in the appendix. Albeit the improvement is not significant due to a short generation length, the consistent improvement brought by cache correction indicates that cache correction effectively mitigates the adverse effects of polluted KV representations, which stem from sparse attention mechanisms and can accumulate errors, potentially reducing model performance.
>
> `Q3. Better performance than full attention baseline`
>
> Besides potential variances, we hypothesize that the rationale behind this phenomenon is that the sparse attention mechanism can focus more on the actual important tokens while removing noisy ones that are irrelevant to the question. We also identify it as a pretty interesting observation but lacks comprehensive analysis, which we leave as a future work.

---

### Official Review · Reviewer_9wNX · 2024-11-03

**Soundness:** 2
**Presentation:** 2
**Contribution:** 2
**Rating:** 5
**Confidence:** 5

**Summary:**

The paper introduce a decoding method for sparse attention with pre-selected tokens. The method essentially performs full attention at lower layers and sparse selective attention at the upper layers to obtain the selected tokens, and then full-attention on reduced sets of tokens in the highest layers. The paper presents experiments to show that TidalDecode is competitive to full attention decoding while reducing latency.

**Strengths:**

- The paper presents an optimization strategy to improve speed and latency by recognizing that the selected tokens in sparse attention remain consistent across layers, so that the selection process may only need to happen in lower layers.
- The paper presents evaluation to show the method is competitive to some baselines on latency evaluation

**Weaknesses:**

- The writing and presentation of the paper significantly need improvement. There are too much unnecessary underlined texts.
- The motivation and introduction are written unclearly, with too much irrelevant introductional texts.
- The preliminary and methodology are written in confusing ways. There is a need to put clearer definition and better writing.

Generally, I would lean towards acceptance if the paper is revised thoroughly.

**Questions:**

NA

---

> ### Author Response · Authors · 2024-11-20
> **Respond to reviewer 9wNX**
>
> We thank the reviewer for their review and address their main concerns below.
>
> `Q1. Writing - underlines`
>
> We thank the reviewer for pointing out the issues in our presentation. In our revised paper, we removed most of the phrases emphasized with underlines and only kept essential ones when they first appeared.
>
> `Q2. Writing - motivation and introduction`
>
> We have removed some unnecessary descriptions in the introduction section and rewritten some paragraphs (in blue) to make it more concise while preserving essential information. The introduction section now follows an order of 1. LLM and long context generation background 2. The bottleneck of long context LLM decoding lies in KV cache memory access (motivation) 3. A brief overview of existing methods 4. A brief introduction of our method (TidalDecode) 5. Evaluation results and summary of contributions.
>
> `Writing - preliminary and methodology`
>
> We have rewritten some descriptions in the methodology section that might cause confusion. Additionally, we have added the necessary explanations and definitions for certain figures and terms, like recall rate. All the modifications are highlighted in blue.
>
> We hope we have addressed most of the writing concerns and misunderstandings raised by the reviewer. Please let us know if there is anything else we can further clarify.

---

### Official Review · Reviewer_oNeu · 2024-11-04

**Soundness:** 3
**Presentation:** 4
**Contribution:** 3
**Rating:** 6
**Confidence:** 5

**Summary:**

This paper introduces TidalAttention, which uses a layer-wise selection approach to choose the top-k keys and values needed in top-k attention, thereby reducing the amount of KV cache requiring I/O.

**Strengths:**

The advantages of this paper are as follows:

1) The logic is very clear, and the writing is well-structured. The figures and captions are precise, making it easy for readers to quickly understand.


2) The focus of the paper is excellent, and I completely agree with it. Particularly with group query attention, KV cache storage is no longer a major issue; instead, the I/O for the KV cache in attention deserves attention. And I really like the mechanism cache correction.


3) Most of the experiments are solid, although there is room for improvement. And the cuda kernel is good.

**Weaknesses:**

I think the paper has the following limitations or raises some questions for me:

1) I believe there may be an issue with the dataset selection in the experiments. Since the authors’ method retains the full KV cache as opposed to KV cache eviction, it would be best to evaluate on datasets with longer outputs. If the dataset only outputs one token, as in NIAH, there should be no difference between eviction-based and selection-based sparse methods, which doesn't highlight the importance of maintaining the full KV cache. Additionally, some QA datasets in LongBench also have this issue. I recommend that the authors supplement more summarization datasets from LongBench (such as gov_report) and code completion datasets (like LCC and repo-bench).


2) The authors claim that their kernel stores dot product instead of attn score, so I just wondering how this method works for group query attention. How can the authors decrease the IO of gqa if the queries in the same group with different top-k?


3) The authors’ description of Quest is not very clear, and it would be beneficial to add citations for previous baselines in the tables.

**Questions:**

see cons.

---

> ### Author Response · Authors · 2024-11-20
> **Respond to reviewer oNeu**
>
> We thank the reviewer for their review and address their main concerns below.
>
> `Q1. dataset selection in the experiments.`
>
> We agree with the reviewer that selecting a dataset with a longer generation length is important to distinguish between the eviction-based and selection-based methods. To this end, we further provide experimental results on the summarization task GovReport and code completion tasks RepoBench (RB) and LCC in Table 5 below and the revised paper. TidalDecode achieves the highest score on the added summarization task GovReport, where the model generates 512 tokens for each question. For code completion tasks, when selecting top-4096 tokens, TidalDecode outperforms full attention and Quest on RepoBench and stays close to full attention on LCC. We also include Table 4 below and in the revised paper to show the generation length for each task in LongBench.
>
> Table 4:
> |        | MFQA | NrtvQA | Qasp | 2Wiki | HotQA | GovRep | QSum | TrivQA | PRe | RB | LCC | Avg |
> |--------|------|--------|------|-------|-------|--------|------|--------|-----|----|-----|-----|
> | Length | 64   | 128    | 128  | 32    | 32    | 512    | 512  | 32     | 32  | 64 | 64  | 145 |
>
> Table 5:
> |                 | GovRep    | RB        | LCC       | Avg       |
> |-----------------|-----------|-----------|-----------|-----------|
> | Full            | 16.77     | 43.97     | 44.88     | 33.11     |
> | Quest (K=1024)  | 15.89     | **49.43** | **49.66** | **31.61** |
> | TD+L13 (K=1024) | **16.41** | 43.29     | 40.67     | 31.49     |
> | Quest (K=4096)  | 16.71     | 43.87     | 43.84     | 32.13     |
> | TD+L13 (K=4096) | **16.78** | **44.54** | **44.27** | **33.50** |
>
> Additionally, some other evidence that shows the improvement brought by TidalDecode: 1. In the original and extended LongBench results (Table 3 and Table 5), TidalDecode can both achieve a better score on average compared to the full-attention version across all eight or eleven tasks, where the average generation length is 145 tokens for a test. 2. In our sensitivity analysis for the choice of the token reselection layer (Table 9-14), choosing different token reselection layers can introduce a huge model performance degradation (e.g., from 100% to 0%), which indicates that the generation quality in the needle-in-the-haystack tasks also highly depends on the algorithm we used in the decoding stage.
>
> `Q2. I/O for group query attention.`
>
> As state-of-the-art kernel implementation for the Group Query Attention (GQA) only loads the keys and values from KVCache once within a group for all the query heads in the same group, if we choose to use a token budget of $k$, then the IO reduction ratio for KV loading would be: $\frac{\min{(\text{seqlen}, \bigcup_{i \in \text{q-head-group}}B_i)}}{\text{seqlen}}$ where $B_i$ is the top-k token indices selected by query head $i$ within the group. So, the reduction rate range spans from $\frac{k}{\text{seqlen}}$ to $\min{(1, \frac{\text{GroupSize}*k}{\text{seqlen}})}$. In practice, using a $k$ satisfying $\text{GroupSize}*k \ll \text{seqlen}$ can already achieve a pretty good performance, and the query head will usually share many top-k indices to reduce the I/O further.
>
> For the dot product results that we saved, the I/O is negligible compared to the I/O for the KVs as the dot product is of shape (batch_size, seq_len, q_head_num) while the KVs are 2*(batch_size, seq_len, head_dim*kv_head_num). In the future, we plan to remove the materialization of the dot product by fusing the top-k operation with the attention kernel.
>
> `Q3. Presentation and references.`
>
> We thank the reviewer for pointing out the issues in our presentation. We have added all the citations for the baselines in the caption of Table 1 and a necessary description of Quest in section 4.2.1. All the modifications are highlighted in blue.

---

> > ### Comment · Reviewer_oNeu · 2024-11-21
> >
> > So basically you do not use GQA in tidal attention, right?

---

> > > ### Author Response · Authors · 2024-11-21
> > > **Respond to reviewer oNeu**
> > >
> > > Most of our evaluation results are on LLaMA-3/3.1/3.2, which uses GQA, so our method can support GQA-based models. However, within the same KV group, instead of using a shared set of top-k indices, tidal attention allows each query head to calculate their top-k indices independently for better performance. But we want to note that we only need to fetch KVs once for overlapping indices among different query heads as they share the same KV head. More specifically, if we use $B_i$ to denote the top-k indices calculated by each query head, we only need to fetch $\bigcup_i{B_i}$ in the sparse layers. We thank the reviewer for the comment and would be happy to address any additional concerns from the reviewer, if any.

---

> > > > ### Comment · Reviewer_oNeu · 2024-11-21
> > > >
> > > > How is your kernel (with attn score) compared with flash_attn_with_kvcache?

---

> > > > > ### Author Response · Authors · 2024-11-21
> > > > > **Respond to reviewer oNeu**
> > > > >
> > > > > For the full attention baseline in the efficiency evaluation section (section 4.3), we are using the state-of-the-art FlashInfer kernel implementation which leverages the flash attention decoding with kvcache. Our kernel implementation also uses the flash attention technique where the attention score is never fully materialized, so that's why our current implementation is storing the dot product between query and keys rather than the attention score. However, to further optimize, we are planning to fuse the argtop-k operator with the attention kernel so there is no need to store the dot product anymore (in a way that each cuda block have a local top-k followed by a reduction over all blocks to gather the final top-k).

---

### Author Response · Authors · 2024-11-20
**General respond**

We thank the reviewers for all their constructive comments. Based on the comments (please refer to our individual replies for details), we have prepared a revised submission, where we focus on clarifying the most important confusions we identified and providing additional results as much as possible. All revised texts are in blue. Most of the additional experimental results and figures are included in Appendix A.1-3.

Summary:
1. Added Figure 8 to the appendix, which depicts the cosine similarity of the attention scores between adjacent layers to further support our motivation to use position persistent sparse attention.
2. Added the generation length in Table 4 and additional results on LongBench in Table 5 in the appendix to demonstrate the effectiveness of our method on tasks with a longer generation length.
3. Added Figure 9 in the appendix to show the effects of KV cache correction.
4. Removed redundant underlines to improve the presentation of the paper.
5. Rewrote the introduction section and the methodology section to make the motivation as well as the description of the methodology clearer to the audience.
6. Added citations to baseline methods and a brief description of Quest.
7. Rewrote the description for Figure 2 to clarify certain terms (recall rate) to resolve potential confusion.

---

### Meta-Review · Area_Chair_121q · 2024-12-19

**Metareview:**

The paper proposes a novel sparse attention mechanism, TidalDecode, to enhance the efficiency and accuracy of large language models (LLMs) during the decoding stage of inference. This mechanism leverages a layer-wise selection approach to optimize attention computation. By reducing the amount of KV cache requiring input/output, TidalDecode significantly decreases latency. Empirical evaluations demonstrate that TidalDecode achieves up to a 2.1× reduction in latency while maintaining comparable generative performance to full attention mechanisms.

The paper is clear and easy to follow. Sufficient experiments and analyses make the results very convincing. In the rebuttal stage, the authors actively interact with reviewers to answer their questions. I am happy to recommend acceptance of the paper. However, please also consider the negative comments when revising the paper.

**Additional Comments On Reviewer Discussion:**

As demonstrated in the 'General Respond' in official comment, the authors have actively refined the paper to address and alleviate the reviewers' concerns.

---

### Decision · Program_Chairs · 2025-01-22

Accept (Poster)